# Molecular Markers of Kidney Transplantation Outcome: Current Omics Tools and Future Developments

**DOI:** 10.3390/ijms23116318

**Published:** 2022-06-05

**Authors:** Maryne Lepoittevin, Thomas Kerforne, Luc Pellerin, Thierry Hauet, Raphael Thuillier

**Affiliations:** 1Inserm Unit Ischémie Reperfusion, Métabolisme et Inflammation Stérile en Transplantation (IRMETIST), UMR U1313, Faculty of Medicine and Pharmacy, University of Poitiers, F-86073 Poitiers, France; maryne.lepoittevin@univ-poitiers.fr (M.L.); thomas.kerforne@chu-poitiers.fr (T.K.); luc.pellerin@univ-poitiers.fr (L.P.); thierry.hauet@univ-poitiers.fr (T.H.); 2Cardio-Thoracic and Vascular Surgery Intensive Care Unit, Coordination of P.M.O.; Centre Hospitalier Universitaire (CHU) Poitiers, F-86021 Poitiers, France; 3Biochemistry Department, CHU Poitiers, F-86021 Poitiers, France; 4University Hospital Federation SUPPORT Tours Poitiers Limoges, F-86021 Poitiers, France

**Keywords:** molecular biomarkers, predictive tool, OMICS, renal transplantation

## Abstract

Purpose of review: The emerging field of molecular predictive medicine is aiming to change the traditional medical approach in renal transplantation. Many studies have explored potential biomarker molecules with predictive properties in renal transplantation, issued from omics research. Herein, we review the biomarker molecules of four technologies (i.e., Genomics, Transcriptomics, Proteomics, and Metabolomics) associated with favorable kidney transplant outcomes. Recent findings: Several panels of molecules have been associated with the outcome that the majority of markers are related to inflammation and immune response; although. other molecular ontologies are also represented, such as proteasome, growth, regeneration, and drug metabolism. Throughout this review, we highlight the lack of properly validated statistical demonstration. Indeed, the most preeminent molecular panels either remain at the limited size study stage or are not confirmed during large-scale studies. At the core of this problem, we identify the methodological shortcomings and propose a comprehensive workflow for discovery and validation of molecular biomarkers that aims to improve the relevance of these tools in the future. Summary: Overall, adopting a patient management through omics approach could bring remarkable improvement to transplantation success. An increased effort and investment between scientists, medical biologists, and clinicians seem to be the path toward a proper solution.

## 1. Introduction

Kidney transplantation represents the optimal therapeutic strategy for patients with end-stage renal disease (ESRD), increasing quality of life [1]. However, despite improvements in immunosuppressive therapy, decreasing acute rejections (AR) rates, and increasing half-life, late kidney graft loss remains a major challenge in transplantation [2].

Nowadays, accepting an organ for transplantation is complex and hazardous. Tools to anticipate outcomes are based on serum creatinine and donor characteristics, including biochemistry, visual appraisal, and time zero biopsies. However, these approaches have drawbacks: serum creatinine is not specific to injury type and is a marker of advanced kidney damage; while renal biopsy, representing the gold standard in transplant evaluation, is too invasive and cannot be performed frequently [3]. Accordingly, clinicians are looking for new molecular markers and fueling research into innovative noninvasive biological markers for early determination and post-transplant success.

Currently, graft evaluation relies on a limited set of molecular biomarkers, often only one parameter, closely correlated with a single functional aspect of the organ involved, or with a specific disease action monitoring. Such molecular markers would ideally be used throughout the transplantation process, from the donor, through organ procurement, preservation, and after transplantation. Donor molecular biomarkers may help in anticipating (short-) and long-term outcomes, while post-transplant, molecular markers may improve knowledge on graft adaptation and its shortcomings [4].

Reviews have been conducted on markers issued from high-throughput omics technologies [4,5]. However, to our knowledge, no review investigates omics methodological challenges and the validity of the derived prediction tools, representing a crucial aspect for the legitimacy of the results.

Herein, we review the principal predictive models available for kidney transplantation and explore the current data on novel innovative tools for before procurement and after transplantation evaluation, discussing their advantages and limits.

## 2. What Are Omics-Based Molecular Biomarkers?

A biomarker is defined as “a biological molecule found in blood, other body fluids, or tissues that can be used to follow body processes and diseases in humans and animals” [6]. In the clinic, biomarkers may be used as a diagnostic, prognostic, classifying, and/or monitoring tool [7].

Nowadays, graft evaluation relies on few biomarkers, and the search for a “golden molecule”, capturing the function of a transplanted organ in all its complexity, has been fruitless. Indeed, the complexity of the mechanisms underlying graft outcome can likely only be captured with a multiparametric molecular approach.

High-throughput technologies have revolutionized medical research. The advent of “omics” technologies implies a comprehensive assessment of a set of molecules. There are four main types of molecular omics: genomics, transcriptomics, proteomics, and metabolomics; each with the potential to improve understanding of pathophysiological mechanisms, support diagnosis, and ultimately lead to new therapeutics and improved outcomes [8]. Genomics is the study of the genome [9], transcriptomics of all RNA molecules transcribed from the genome [10], proteomics of protein nature, and structure, function [11], and metabolomics of metabolites and their related chemical processes [12]. The metabolome is a particular challenge since it is a dynamic system constantly in flux.

To date, the use of these approaches in renal transplantation has been mainly restricted to research, and transition to clinical practices remains a challenge. Indeed, molecular biomarker validation, identification, or verification is laborious and demands particular attention. The tools involved in exploring these biomarkers are also to be taken into consideration; indeed, as we will expose, in many publications, the validity of biomarkers is assumed where it should be evaluated and re-evaluated.

## 3. Molecular Omics before Procurement

The need for organs greatly exceeds donations, leading professionals to use extended criteria donor organs, more sensitive to ischemia/reperfusion injury, unavoidable during transplantation, and associated with adverse outcomes.

Anticipating organ performance is a serious clinical challenge, which can improve allocation and individualize post-transplant management. To this end, several predictive models have been developed [13] (Table 1).

The leading algorithm is the Kidney Donor Profile Index (KDPI), currently the most effective scoring system to quantify graft failure risk. It includes ten variables: age, height, weight, last serum creatinine, history of diabetes, hypertension, HCV-infection, ethnicity, and the cause of death [14]. It should be noted that this score is not intended to serve on its own, as the predictive power of the KDPI is moderate (c-statistic = 0.60) and it does not take into consideration factors that may impact graft outcomes, such as any damage, injury, or abnormalities of the donor [15]. Further studies are needed to determine the accuracy of predictive tools based on common clinical parameters and to compare or improve their performance with more sophisticated biomarkers using omics.

In this section, we will review studies performed using omics and their prediction models based on “before procurement” data (i.e., reference to the donor and recipient biomarkers before transplantation).

### 3.1. Genomics

Despite early enthusiasm in demonstrating a potential association between donor DNA polymorphisms and outcome, no study could show good predictive performance. The only exception is the *APOL1* polymorphism, a molecule involved in the formation of most cholesteryl esters in plasma that also promotes the efflux of cholesterol from cells, and has been validated technologically and statistically (*p* < 0.0001) in large single- and multi-center studies. It is mainly expressed in African ancestry populations and associated with worse outcomes [16,17]. It is now accepted in the medical community and integrated in the KDPI.

Other studies focused on gene coding for immune-related molecules, exploring the link between late allograft loss and genetic variation in the immune response. Authors have demonstrated that genotypic polymorphisms of *TGF*-β and *CCR5* genes were associated with acute rejection (AR) [18]. However, despite numerous publications showing plausible polymorphisms associated with AR, there are no consistent genetic predictors of acute kidney allograft rejection [27]. Others have investigated genetic effects on deceased donor/recipient. Despite being one of the largest cohorts in transplantation, no gene variants have shown robustness equal or superior to the *HLA* gene [28].

### 3.2. Transcriptomics

Transcriptomics quickly followed genomics [29], using high-throughput techniques such as microarray and next-generation sequencing [30]. However, few studies have been conducted on donors.

A genome-wide gene expression study on human donor kidney studied 48 genes coding for molecules linked to cell communication, apoptosis, inflammation, and immune response. The authors show that these were increased in cadaveric donors vs. living donors and increased the risk of acute renal failure [19]. An unsupervised analysis in donor kidney biopsies reported a molecular panel of 1051 transcripts differentially expressed between deceased vs. living donors, as the former over-expressed molecules were related to inflammation (immunoglobulins), collagens, integrins, chemokines, Toll-like receptor signaling, antigen processing and presentation, and renal injury; while the under-expressing markers were transport, glucose, and fatty acid and amino acid metabolism [20]. This was later refined to 36 molecular candidates in deceased-donor kidney biopsies: *IGFBP5* and *CSNK2A2* (cell cycle/growth); *RASGRP3* (signal transduction); *CD83, BCL3, MX1*, and *TNFRSF1B* (immune response); *ENPP4, GBA3* (metabolism), which was significantly associated with the stratification of graft performance in correlation with the recipient’s DGF (*p* < 0.001) [21]. Later, the same authors used several molecules to stratify graft performance, using a panel associated with antigen processing and presentation via MHC class I/II, T-cell-mediated cytotoxicity, allograft rejection/graft vs. host disease, antigen processing and presentation, and cell adhesion molecules (top molecules were *HLA-G, HLA-E, HLA-DRB1, HLA-DRA, HLA-DPB1, HLA-DPA1, HLA-DQB1, HLA-DQA1, HLA-B, HLA-C, HLA-DMA, PSMB8, PSME1, HSP90AB1, and PRDX1*). This molecular panel was able to discriminate kidneys that would later develop DGF and impaired 1-month function [22].

Recently, a 23-transcript molecular signature associated with NK and CD8+ T-cell activation (among which Granzyme B, FGFBP2, NKG7, Perforin 1, Fas Ligand, CD8A, and CCR5coagulation factor XII) was described in pretransplant blood to predict AR [23] with encouraging results (AUC = 0.89); however, it was in a limited size study (*n* = 80) and will need to be confirmed in large transplant cohorts.

### 3.3. Proteomics

Several proteomic studies have been carried out in recent years, mostly focusing on urine and perfusate [31], aiming to define organ quality through a holistic approach integrating the multiple events of donor kidney injuries. Comparing proteomic profiles between different pathophysiological conditions highlighted protein panels with interesting predictive performance for graft failure, but always in a limited size study (*n* = 113) [5,24]. Among these, the use of Neutrophil gelatinase-associated lipocalin (NGAL) and L-type fatty acid-binding protein (L-FABP) showed good performance in the prediction of reduced graft function (AUC 0.8); however, this remains at the limited size study (*n* = 94) stage.

### 3.4. Metabolomics

Metabolomics is now acknowledged as a potential approach in transplantation, generally used for molecular biomarkers discovery and generating biological hypotheses. One such study used 266 plasma molecular metabolites to build ANOVA multiblock OPLS models; the main molecules being azelaic acid, creatinine, kynurenic acid, kynurenine, indoxyl sulfate, and tryptophan. They showed a strong association with rejection prediction (*p* < 0.05) [25].

Recently, a review was conducted on metabolomics investigation during perfusion for the heart, lung, kidney, and liver. Biomarker molecules mainly associated with energy metabolism (ATP → Pi, Krebs cycle intermediates, lactate), glycogenolysis, and amino acid metabolism were discovered. There is an interesting association with graft quality, warranting larger studies [26]. Indeed, perfusate is a non-invasive alternative to biopsies, enabling frequent sampling and confirming the superiority of hypothermic machine preservation over traditional preservation.

To date, these multi-omics molecular investigations coalesce into a large panel of targets, but without coherence. One of the main challenges stems from the absence of consensus or a centralized database, inducing sensitivity and specificity for diagnostic analysis. Development of uniformized workflow for exploration and predictive model building is needed. This would open the way toward organ-tailored preservation, whereby high-risk grafts undergo an assessment by omics-conducted molecular profiling, leading to re-conditioning before transplantation [26].

## 4. Molecular Omics after Transplantation

Kidney biopsy and creatinine are the main graft monitoring molecules agreed upon in the transplantation community. However, they cannot apprehend the complete phenotype of a patient. Hence, machine learning-based prediction algorithms have been explored. The iBox prognostication system was developed, estimating long-term allograft survival. iBox defines a score based on immunologic, histologic, and functional (eGFR and proteinuria) recipient’s criteria [32]. Other works have also been described with common molecules but different classifier approaches [33,34]. Although such allograft survival prediction models hold promise, these tools work well at the population level but lose accuracy for a specific individual [35].

Omics molecular profiling provides new resources to improve prediction. This section aims to overview studies performed after transplantation (Table 2).

### 4.1. Genomics

Genomic studies have become of great importance in kidney transplantations [5,44]. The first landmark genomics study uncovered for the first-time a molecular heterogeneity in AR [45].

Recently, genomic analyses have given more specific and robust results. Authors investigated a panel of 13 genes: *MET*, *ST5*, and *KAAG1* (tumor development or suppression); *RNF149, ASB15*, and *KLH13* (ubiquitination and proteasome); *TGIF1, SPRY4, WNT9A, RXRA*, and *FJX1* (developmental or growth pathways such as *NOTCH/Wnt* or *RAR*); and *CHCHD10* and *SERINC5* (energy and membrane repair). They demonstrated good predictive power for the development of fibrosis at 1 year (AUC 0.9) [36]. Researchers have mobilized to unify genomics databases and predictive model selections, allowing prediction of allograft function, late allograft failure, or tolerance [46]; with for instance the demonstration of an association with long-term allograft function (*p* = 0.004) for polymorphisms of several genes such as *CYP3A5* (involved in drug metabolization), *CCR5*, *FOXP3*, and other genes involved in inflammation and immune response (interleukins, chemokines, TLR pathway, and innate and adaptative immunity mediators); *TGF b, VEGF*, and other mediators of fibrosis. However, this was performed in a small subset and requires larger studies, meta-analysis, and subsequent validation [37].

### 4.2. Transcriptomics

Almost two decades ago, it was demonstrated that recipient urinary cell mRNA screening has similar diagnostic performance as biopsy histological analysis for AR [38]. Later, these transcript sets were shown to correlate with biopsy diagnosis, themselves having limitations for some diagnoses of rejection, identifying some of the limitations of Banff’s classification [47]. Such transcriptomics data demonstrated unique findings in clinical settings, whether in recipients’ peripheral blood, urine, or biopsies.

Further investigation on peripheral blood transcriptomics molecular analysis highlighted a 17-gene set to detect high risk of AR with high sensitivity (AUC = 0.93). The target molecules were *DUSP1, CFLAR, ITGAX, NAMPT, MAPK9, PSEN1, RYBP, NKTR, SLC25A37, CEACAM4, RARA, RXRA, EPOR, GZMK*, and *RHEB*) together with 18S ribosomal RNA as a housekeeping gene, a molecular signature mainly directed at defining the type and intensity of the inflammatory response. This project, called the Kidney Solid Organ Response Test (kSORT), is currently undergoing validation in prospective clinical trials [40,41]. However, an independent study showed that adding kSORT to classical clinical variables (eGFR, Proteinuria, and DSA) did not increase their diagnostic performance [39].

Recently, a manuscript described the design and methodology of a new clinical trial that investigated three genes: 18S-normalized *CD3ε, CXCL10* mRNA, and 18S ribosomal RNA, all associated with inflammation and immune response, to determine their predictive potential for rejection and infection in pediatric kidney transplant recipients: the VIRTUUS molecular panel [42].

### 4.3. Proteomics

Proteomics investigations of molecular biomarkers have been increasingly used post-transplantation. The most promising explored urinary levels of *CXCL9* and *CXCL10* proteins, both linked to inflammation signaling, were able to show good predictive potential for T-cell-mediated rejection (TCMR) and antibody-mediated rejection (ABMR) (AUC: 0.75 and 0.83, respectively). However, none have been implemented in clinical practice [43].

Furthermore, smaller studies found differing patterns of protein biomarkers associated with short- and long-term outcomes; however, there is no uniformly agreed molecular panel as each study; similar to before-procurement studies, has different disease definitions, sample collection, and methodologies for data acquisition and analysis [5,48].

### 4.4. Metabolomics

Studies explored systemic metabolism-related molecular changes after kidney transplantation in serum, plasma, and urine, all with the goal of reflecting the key processes related to graft accommodation and possibly predicting rejection [25,49,50].

Overall, none of these studies resulted in molecular biomarkers progressing beyond the discovery stage for similar reasons to proteomics; that is, a lack of a more unambiguous identification of metabolite biomarkers and extensive validation efforts to enable these markers to be integrated into patient health care [51].

## 5. Toward an Optimized Use of Omics in Clinical Application: Workflow, Advantages, and Limits

Altogether, these omics approaches could become valuable tools in the clinic to discover and validate new molecular biomarkers and to open the possibility of preservation tailored to the needs of the organ, increasing the number and the quality of donor organs. Rather than employing a reductionist approach, these systems use a holistic and integrative approach to better capture ongoing biological processes and their related molecules.

However, integration of such technologies and molecules in clinical practice requires an understanding of the dynamics underlying molecular biomarker discovery and model development [51,52,53]. Hence, we propose to summarize these into a specific workflow, as is presented in Figure 1.

The first phase is the DISCOVERY. It focuses on hypothesis generation without a priori and is generally untargeted [51]. In short, this approach will extract the relevant molecular markers from the pool, rather than start from hypothetically key molecules and attempt to verify their importance.

Figure 1, a: *Experimental design*, defining targeted clinical issues, molecule types that are likely involved (metabolites, nucleic acids, proteins, etc.), and thus the choice of high-throughput technology to use.

Figure 1, b: *Defining the molecular investigation technology*, next-generation sequencing, microarray, or mass-spectrometry, the major omics platforms have distinct limitations in sampling and instrumental methods that must be taken into account. Human sample collection, storage, and preparation are also critical issues, aiming at limiting internal degradation of target molecules while remaining compatible with a clinical setting. This may involve the use of stabilizer reagents (for instance, with RNA) or of pre-analytical steps such as deprotonation (for instance, with metabolomics). In terms of the instrumental method, optimal parameters selection is laborious: on the one hand, the industrial sector offers a large catalog of these tools; on the other hand, the adjustment is conditioned by the (often partial) researcher’s knowledge and the literature. To date, no consensus defines an efficient configuration [50]; hence, the need for lab-specific optimization is unavoidable.

Figure 1, c: *Data processing*, the instrument generates a large dataset containing hundreds to thousands of molecules and their respective detected level in each sample, producing enormous tables, so-called «Big Data». The process and management of these datasets are complex. The common stages of omics data processing are transformation, normalization, quality control correction, noise filtration, and imputation. All of them are essential and could be a potential source of error. Hence, it requires numerous adjustments and perfect understanding [51,52]. Nowadays, automation is made possible by several types of software, and it promotes some level of standards. However, such solutions, especially commercial ones, sometimes act as «black-boxes», lacking transparency, which could negatively impact future results. These standard approaches lead to «one size fits all» for our data affecting the quality [54]. In most cases, each hypothesis and/or dataset is unique and answers to its own standard approaches. It is therefore necessary to keep a “made to measure” approach while investing significant energy to perform proper controls. Moreover, it is of primary importance to use molecular markers for normalization, such as a housekeeping gene in transcriptomics, but also a marked molecule for mass spectrometry approaches.

Figure 1, d: *Data analysis*, chemometric is a discipline that manipulates data utilizing mathematics and statistic fundamentals [55]. This step is sensitive, as it is easy to get lost in the wide array of tools available [56]. In order to extract the relevant molecules from the available pool present in the Big Data, data science provides a large number of statistical models (also known as classifiers), from typical regressions to advanced machine learning techniques (neural networks, support vector machines, decision trees, logistic and multinomial regression, etc.). No clear guideline seems to exist in transplantation; hence, each team applies their approach and, sometimes involuntarily, chooses the method to their advantage. This is one of the biggest biases in data science: knowing how to properly use modulization on the dataset [34,56,57]. This may bias the data and highlight molecules as important, which will, later on, be discovered as artifacts, leading to loss of time and avoidable spending. Moreover, the accuracy might be of limited significance if a reduced cohort is used, as it could achieve significantly worse results when the method is extended, and reciprocally. There are, however, several seminal works that can help in the building of a more robust modeling strategy [55,58]. The “TRIPOD” guideline promotes some level of standards or tasks to achieve to develop, validate, or update outcomes of statistical models [56].

The second phase is the VALIDATION, aiming at validating the predictive model, and thus the set of molecular biomarker(s) identified in the previous phase, over a large, and new, sample set.

Figure 1, e: *Predicting model and validation*, does the model performance reflects the distance between the predicted and observed outcome? Is the model able to distinguish patients who will experience the event from patients who will not? Does the predicting tool account for competing risks in the population of interest? What is the model power? The performances of the models are usually measured using the concordance statistics that correspond to the ROC AUC for classification and to the correlation coefficient for regression, which take into account the error risks. It is recommended to evaluate the predictive tool both through internal and external validations. Internal validation includes random split-sampling of cohort and resampling methods. External validation includes large uni- and multi-centric cohorts to generalize the tool. Finally, molecular biomarker(s) candidate(s) and these predictive models’ implementation in clinical practice can be considered [51,56].

Figure 1, f: *Workflow optimization* is aimed at the practical adaptation of predictive tools toward easy routine implementation “at the patient’s bedside”. It is important to consider that the most sophisticated, sensitive, and specific molecular biomarker(s) will only reluctantly be accepted in clinical practice if the tool is too complex to implement. To increase speed, it is necessary to re-optimize the analytical approach with only the specific molecules in mind, in order to only target these and maximize executions of the signal acquisition and analysis. This may include for instance-specific buffers and/or adjuvants, which would maximize molecular recovery. Successful implementation into clinical practice hinges on: (i) cost-effectiveness; (ii) availability of clinically realistic sample collection and procedures; (iii) development and validation of viable bioanalytical strategies; and (iv) software tools that allow for data analysis and fast translation into clinically meaningful information [33,34,57].

Figure 1, g: *Integration at the patient’s bedside*, there are two aspects to consider in order to foster the introduction of multiparametric omics-based tools to the clinic: the analytical technology, and the end-user perspective. Regarding technology, omics relies on high-end analytical automata, requiring a high level of expertise. The enrolment of the medical biology services is thus of capital importance, and thankfully such high-end technologies such as mass spectrometry and next-generation sequencing are now well installed in the labs. It is thus possible to envision sending the patient’s sample through the regular bioanalytical route and specialized personnel to perform the molecular extraction and run of the sample, with the appropriate internal and external controls. Regarding the end-user perspective, this is where the versatility of machine learning classifiers can be an asset; while the diversity of possible outputs and overly complex molecular names can be overwhelming at first, the involvement of a medical biologist trained in biology and data science can transform it into a result tailored to the clinician’s need (on the donor of the recipient side), with either a score, a decision tree, a relative risk, or a combination of such. Help from software engineers can also be a bonus to implement the prediction algorithms on user-friendly platforms (apps, webpages, etc.).

It is hoped that this will give the reader insight into the direction, in which the medical sciences should, in our opinion, be moving over the next few years. The study of disease at the molecular level could enable us to treat many diseases more rationally; however, it is indispensable to undertake it the right way, to maximize benefits, and to avoid wasting valuable patient samples. However, this will not happen overnight; there is likely to be a long period of development and evaluation of our new technology before its clinical application can be fully assessed.

## 6. Conclusions

In summary, we address the important issue of predicting the outcome of kidney transplants, which is becoming increasingly relevant in clinical practice. As with other techniques and tools, practical implementation of a molecular biomarker model should be performed with caution and a degree of skepticism. An increased effort and investment between scientists, medical biologists, and clinicians seem to be the path toward a proper solution.

We perceived in renal transplant literature a willingness to work together. However, in the studies reviewed herein, we found substantial variability in data collection, signal processing methods, and choice of classifier to build predictive models. Moreover, the definition of graft failure was particularly variable, and the endpoint may vary depending on the clinician [59]. About half of the studies validated their model without external data. For the few studies with external validation, the performance of their model was not persuasive [51,60].

Even though we paint a negative picture of this field, omics molecular technologies have enormous potential. Among these, metabolomics offers a low-cost option Moreover, sample and data analysis can be performed in a matter of hours, which is compatible with the timing in transplantations. The increased availability of high-definition equipment in hospital biochemistry services (such as mass spectrometer, next-generation sequencing machines, etc.) can foster the use of many omics technologies toward improving transplantation, keeping the itinerary of the sample within typical lanes in medical practice, hence improving the possibility for deployment into routine use [57,61]. Although our study focused only on the four fundamental pillars, other omics technologies exist. These other omics, including epigenomic, fluxomic, inomic, or lipidomic, could bring a novel perspective to the field of transplantation [62,63].

As discussed above, the first promising steps have been made and clinical molecular markers in combination with predictive models could provide valuable information on allograft outcomes. In the future, omics technologies could bring a new regard to current clinical practice and patient/graft management (Figure 2).

## Figures and Tables

**Figure 1 ijms-23-06318-f001:**
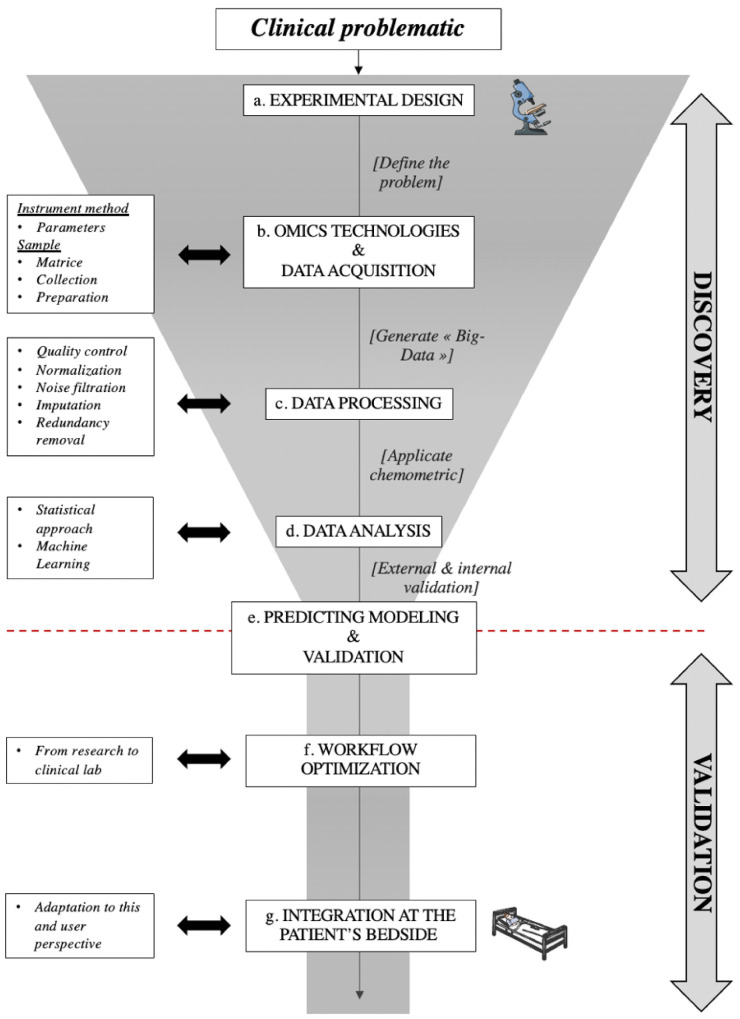
Toward an optimized use of omics in clinical application: workflow, advantages, and limits. The workflow is divided into two sections: Discovery (I) and Validation (II), which in turn are divided into several steps. All steps are described in the section, “Toward an optimized use of omics in clinical application: workflow, advantages, and limits” of the review.

**Figure 2 ijms-23-06318-f002:**
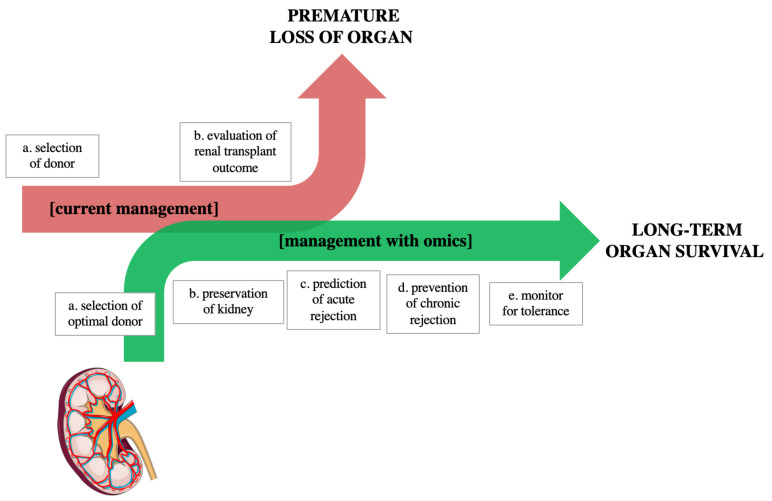
New perspective on renal transplant management. The arrows show the current management (red arrow) and future potential management (green arrow) adopted for kidney transplantation. Current management includes two steps: (**a**) Donor–Recipient pairing, based on evaluation of donor variables and organ quality. To help clinicians in allocation, the Kidney Donor Profile Index (KDPI) score can be used. (**b**) After kidney transplantation monitoring, using kidney biopsy and creatinine level. However, such parameters cannot apprehend the complete phenotype of a patient and can lead to premature loss of organs. Future management with omics: to improve quality of care in patients, omic approaches (i.e., genomics, transcriptomics, proteomics, and metabolomics) can be implemented through several steps: (**a**) selection of optimal donor with the addition of omics-based predictive tools to common clinical parameters; (**b**) preservation of kidney, which can be tailored to the need of the organ based on an in-depth understanding of the organ (through omics data at the donor level) and through real-time monitoring of the perfusate; (**c**) post-transplant monitoring and prediction of acute rejection; throughout the life of the organ, graft management through bioinformatics could improve clinical practice; and (**d**) prevent chronic graft loss; and (**e**) monitor for tolerance. Figure adapted from reference [64].

**Table 1 ijms-23-06318-t001:** Pre-transplantation molecular biomarkers; their role and the performance as predictive tools in transplantation.

Predictive Model Approach	Markers, Molecules, Roles	Sample Type	Performance	Limitation	Ref.
**KDPI**	Age, height, weight, last serum creatinine, history of diabetes, hypertension, HCV-infection, ethnicity, and the cause of death	Blood	Prediction of graft failure (AUC 0.6)	Not validated in European cohorts, low c statistics	[14,15]
**Genomics**	*APOL1* polymorphism, involved in the formation of most cholesteryl esters in plasma and also promotes efflux of cholesterol from cells	Blood	Significantly associated with worse outcome (*p* < 0.0001), now integrated to KDPI	Limited to patient of african descent	[16,17]
	Polymorphisms of *TGF-**β* and *CCR5*, role in inflammation	Blood	no consistent association with acute rejection	Small cohorts	[18]
**Transcriptomics**	48 mRNA coding for cell communication, apoptosis, inflammation	Biospy	correlation with risk of graft failure	Limited number of samples	[19]
	Molecular pannel of 1051 transcripts; overexpression of molecules related to inflammation (immunoglobulins), collagens, integrins, chemokines, Toll-like receptor signaling, antigen processing and presentation and renal injury; underexpression of markers of transport, glucose, fatty acid and amino acid metabolism	Biospy	Many molecules differentiated between organs from deceased donors vs. living donors (adjusted *p*-value <0.01)	Small cohorts and short duration of follow-up	[20]
	36 candidate genes, chief among which *IGFBP5* and *CSNK2A2* (cell cycle/growth); *RASGRP3* (signal transduction); *CD83, BCL3, MX1, TNFRSF1B* (immune response); *ENPP4, GBA3* (metabolism)	Biospy	Significantly associated with stratification of graft performance in correlation with recipient’s DGF (*p* < 0.001)	Small cohorts	[21]
	Molecular pannel associated with antigen processing and presentation via MHC class I/II, T-cell–mediated cytotoxicity, allograft rejection/graft versus host disease, antigen processing and presentation and cell adhesion molecules. Top molecules were *HLA-G, HLA-E, HLA-DRB1, HLA-DRA, HLA-DPB1, HLA-DPA1*, *HLA-DQB1, HLA-DQA1, HLA-B, HLA-C, HLA-DMA, PSMB8, PSME1, HSP90AB1*, and *PRDX1*	Biospy	Significantly associated with DGF severity (*p* < 0.001)	Small cohorts	[22]
	23-gene transcriptional signature associated with NK and CD8+ T cell activation, among which *Granzyme B, FGFBP2, NKG7, Perforin 1, Fas Ligand, CD8A, CCR5*, coagulation factor XII	Blood	Risk score associated with acute cellular rejection after 6 months, antibody-mediated rejection and/or de novo donor-specific antibodies, and graft loss (AUC 0.89)	No standardization	[23]
**Proteomics**	Predictive model using Neutrophil gelatinase-associated lipocalin (*NGAL*) and L-type fatty acid binding protein (*L-FABP*)	Urine	Prediction of reduced graft function (AUC 0.8)	Small cohorts	[24]
**Metabolomics**	266 plasma metabolites building ANOVA multiblock OPLS models, the main molecules being azelaic acid, creatinine, kynurenic acid, kynurenine, indoxyl sulfate and tryptophan	Blood	Significantly associated with rejection (*p* < 0.005)	Data interpretation and small cohorts	[25]
	Review on metabolomics investigation during perfusion for the heart, lung, kidney and liver. Biomarkers molecules mainly associated with energy metabolim (ATP → Pi, Krebs cycle intermediates, lactate), glycogenolysis, amino acids metabolism,		Measurable association with graft quality	Small cohorts	[26]

**Table 2 ijms-23-06318-t002:** Post-transplantation molecular biomarkers; their role and the performance as predictive tools in transplantation.

Predictive model approach	Markers, Molecules, Roles	Sample Type	Performance	Limitation	Ref
Genomics	Pannel of 13 genes : *MET, ST5* and *KAAG1* (tumor development or suppression); *RNF149, ASB15, KLH13* (ubiquitination and proteasome) ; *TGIF1, SPRY4, WNT9A, RXRA* and *FJX1* (developmental or growth pathways such as *NOTCH/Wnt* or *RAR*); *CHCHD10* and *SERINC5* (energy and membrane repair)	Biopsy	Prediction of the development of fibrosis at 1 year (AUC 0.9)	No validation yet, clinical trial ongoing	[36]
	Polymorphism of several genes such as *CYP3A5* (involved in drug metabolisation, among which tacrolimus), *CCR5*, *FOXP3* and other genes involved in inflammation and immune response (interleukines, chemokines, TLR pathway, innate and adaptative immunity mediators); *TGF b, VEGF* and other mediators of fibrosis.	Biopsy	Several variants are predictors of long-term allograft function (*p* = 0.004)	Very small sample set (24 specimens)	[37]
Transcriptomics	Non-invasive urinary cell mRNAs *Granzyme B, Perforin, Cyclophilin B*, all related to the immune system and inflammation	Urine	Significantly associated with acute rejection (*p* < 0.001)	Small cohort	[38]
	The kSORT pannel: 17-gene transcriptional signature to predict acute rejection *DUSP1, CFLAR, ITGAX, NAMPT, MAPK9, PSEN1, RYBP, NKTR, SLC25A37, CEACAM4, RARA, RXRA, EPOR, GZMK,* and *RHEB*) together with 18S ribosomal RNA as housekeeping gene. This signature is mainly directed at defining the type and intensity of the inflammatory response	Blood	Prediction of Acute Rejection (AUC = 0.93)	No validation on an independent sample set. Indeed, an independant study showed that adding kSORT to classical clinical variables (eGFR, Proteinuria, DSA) did not increase their diagnostic performance [39]	[40,41]
	The VIRTUUS panel: 3 genes (18S-normalized *CD3**ε, CXCL10* mRNA, and 18S ribosomal RNA) associated with inflammation and immune response	Blood	No result yet	This is a design & method presentation of an ongoing clinical trial	[42]
Proteomics	Urinary levels of *CXCL9* and *CXCL10* proteins, both linked to inflammation signaling	Urine	Prediction of T cell-mediated rejection (TCMR) and antibody-mediated rejection (ABMR) (AUC: 0.75 and 0.83 respectively)	Prospective cohort study	[43]
Metabolomics	None significant studies				

## Data Availability

Not applicable.

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
