# Peer review of "Molecular Markers of Kidney Transplantation Outcome: Current Omics Tools and Future Developments"

_ijms, 2022, doi:10.3390/ijms23116318_

Round 1

Reviewer 1 Report

The authors have provided a comprehensive review of the literature to attempt to consolidate the current understanding and limitations to the use of molecular biomarkers to enable clinicians and doctors to improve outcomes in kidney transplantation events.

The review is comprehensive and understandable to a wide audience and I appreciate the efforts of the authors to summarize the predictive models in the literature for both their performance and their limitations.  Promoting a more optimal workflow organization is one thing but as the authors have pointed out where the weaknesses in the literature (eg. small cohorts, lack of standardization) existed, why not also promote some level of standards or benchmarks to achieve for improved outcomes?  What are the levels to achieve and how to promote the disparate groups working together to achieve better outcomes.  This is much more difficult of course but I feel like at least for the "Discovery Phase" component of your workflow you could suggest what levels need to be achieved.  What entities or organizations make sense to establish these guidelines?

The authors considered four "omics" technologies, and I was left wondering whether "epigenomics" was considered or just lumped into "genomics".  I would assume this "omics" category would have large implications for or even be predictive of transplant success?  Toward the end of the discussion it became clear the authors were promoting the use of "metabolomics" over the other categories primarily for costs and speed, but does the quality of the data and its predictive merit not supposed to be part of the comparisons and the integration of all the potential data?

Most of my remaining comments are minor and intended to help improve clarity of the information as presented.

Line          Comment

16     why use "families", perhaps "sources of information" or "technologies"

18     suggest "... associated with favorable kidney transplant outcomes."    

36     "hazardous"

40     suggest "... biopsy, represents the gold standard ..."

43     suggest "... post-transplant success."

53     suggest "...tools, representing a crucial..."

78     demands

81     re-evaluated

94    What is "c =0,60" ??

108   populations

116   suggest "... no gene variants showed"

125  What is "unsupervised"?

127  over-expressed

129   under-expressing (remove "f")

133   significantly

136   panel

140   panel 

159   biomarker

160   Suggest "One such study used 266 ..."

165-167  Incomplete sentence, "... metabolism were discovered??"

168    Start new sentence with "Indeed perfusate is ..."

169-170 Incomplete sentence

173     "... or a centralized database,... for diagnostic analysis."

176     "... profiling leading to re-conditioning ..."

178     Make heading consistent with Section 3.  "Molecular Omics"

196     panel

204    "among which tacrolimus)" ???

225     suggest "...that investigated 3 genes:..."

229     panel

235     Suggest new sentence start with "However, none has been ..."

238     panel

242     suggest delete "was"

244     suggest "... these studies resulted in ..."

263     Incomplete sentence ?

276     terms

277     parameter

283    data sets

287    "many softwares." ???

361-363   I would try to include this particular thought into the workflow diagram of Figure 1 !

372     suggest delete "an"

526     reference 52 is not in the text

533     reference 55 is not in the text

Table 1.  I would suggest somehow make the lines separating the different approaches more distinct, darker or thicker in some way.  In the column under "Performance" for polymorphisms of TGF-B indicates "no consistent association with acute" acute what? perhaps acute rejection ?

Suggest overall Table 1 name heading to delete "their use"

Table 2.  I would suggest somehow make the lines separating the different approaches more distinct, darker or thicker in some way.  Second column "VIRTUUS panel".  Third column "Prediction of Acute Rejection".  Fourth column spelling of "independent" and "Indeed"

Suggest overall Table 2 name heading to delete "their use"

Author Response

Dear Reviewer,

Thank you for your helpful comments and inputs. We have amended the papers accordingly, and below we list the different changes in response to your comments.

The authors have provided a comprehensive review of the literature to attempt to consolidate the current understanding and limitations to the use of molecular biomarkers to enable clinicians and doctors to improve outcomes in kidney transplantation events.

The review is comprehensive and understandable to a wide audience and I appreciate the efforts of the authors to summarize the predictive models in the literature for both their performance and their limitations.  Promoting a more optimal workflow organization is one thing but as the authors have pointed out where the weaknesses in the literature (eg. small cohorts, lack of standardization) existed, why not also promote some level of standards or benchmarks to achieve for improved outcomes?  What are the levels to achieve and how to promote the disparate groups working together to achieve better outcomes.  This is much more difficult of course but I feel like at least for the "Discovery Phase" component of your workflow you could suggest what levels need to be achieved.  What entities or organizations make sense to establish these guidelines?

Thank you for your comments.

To our knowledge, only one paper promotes some level of standards or task to achieve for improved outcomes. The paper « Guideline TRIPOD » (https://doi.org/10.1186/s12916-014-0241-z) provides recommendations for the reporting of studies developing, validating, or updating a prediction model study and consists of a 22-item checklist detailing the essential information that should be included in a report of a prediction model study. However this is focused on the statistical aspect.

We mentioned this work in the previous version, however we strengthened this in the revision this information. Line:275-276

Regarding the size of the cohort, to our knowledge no strict rule exists. The consensus of data scientists is that 10 patients are needed per parameter, which means that if an algorithm selects Y relevant parameters, the ideal process is to have Y x 10 patients.

For the data processing step, including normalization, there is no document that can propose a standard approach. This is because each hypothesis and/or data set is unique. Having a guideline could be detrimental. It would become a "one size fits all" for the data, which would strongly impact on everyone's results. This step must remain « made to measure » for our data. However, I agree with you that visual and comprehensible controls should be put in place for all.

We clarified this information. Lines:256-262

The authors considered four "omics" technologies, and I was left wondering whether "epigenomics" was considered or just lumped into "genomics".  I would assume this "omics" category would have large implications for or even be predictive of transplant success?  Toward the end of the discussion it became clear the authors were promoting the use of "metabolomics" over the other categories primarily for costs and speed, but does the quality of the data and its predictive merit not supposed to be part of the comparisons and the integration of all the potential data?

Thank you for the relevance of your remarks.

We chose to focus on the main 4 of Omic technologies (Genomics, Transcriptomics, Proteomics and Metabolomics) because these technics have been known to the community for the longest time. Moreover, out main point was the challenge of the implementation in medicine, which we didn’t want to dilute by including too many different approaches such as epigenomic, fluxomic, lipidomic, ionomic or phenomic. We agree that undeed they are significant new branches in the medical field, notably in transplant success and hence could be the subject of another review.

We however added a few references to help the reader widen its focus of interest. Lines:337-339

Regarding your last point, we have clarified the discussion and have provided a more comprehensive conclusion. Line:332-337

Most of my remaining comments are minor and intended to help improve clarity of the information as presented.

Thank you for these minor remarks. We did the modifications on the text. Regarding the questions, you will find our answers below.

Line          Comment

16     why use "families", perhaps "sources of information" or "technologies"

18     suggest "... associated with favorable kidney transplant outcomes."    

36     "hazardous"

40     suggest "... biopsy, represents the gold standard ..."

43     suggest "... post-transplant success."

53     suggest "...tools, representing a crucial..."

78     demands

81     re-evaluated

94    What is "c =0,60" ??

The C-statistic (sometimes called the “concordance” statistic or C-index) is a measure of goodness of fit for binary outcomes in a logistic regression model. In clinical studies, the C-statistic gives the probability a randomly selected patient who experienced an event (e.g. a disease or condition) had a higher risk score than a patient who had not experienced the event. It is equal to the area under the Receiver Operating Characteristic (ROC) curve and ranges from 0.5 to 1

108   populations

116   suggest "... no gene variants showed"

125  What is "unsupervised"?

Unsupervised learning refers to the use of artificial intelligence (AI) algorithms to identify patterns in data sets containing data points that are neither classified nor labeled. The algorithms are thus allowed to classify, label and/or group the data points contained within the data sets without having any external guidance in performing that task. In other words, unsupervised learning allows the system to identify patterns within data sets on its own. In unsupervised learning, an AI system will group unsorted information according to similarities and differences even though there are no categories provided.

127  over-expressed

129   under-expressing (remove "f")

133   significantly

136   panel

140   panel 

159   biomarker

160   Suggest "One such study used 266 ..."

165-167  Incomplete sentence, "... metabolism were discovered??"

168    Start new sentence with "Indeed perfusate is ..."

169-170 Incomplete sentence

173     "... or a centralized database,... for diagnostic analysis."

176     "... profiling leading to re-conditioning ..."

178     Make heading consistent with Section 3.  "Molecular Omics"

196     panel

204    "among which tacrolimus)" ???

225     suggest "...that investigated 3 genes:..."

229     panel

235     Suggest new sentence start with "However, none has been ..."

238     panel

242     suggest delete "was"

244     suggest "... these studies resulted in ..."

263     Incomplete sentence ?

276     terms

277     parameter

283    data sets

287    "many softwares." ???

361-363   I would try to include this particular thought into the workflow diagram of Figure 1 !

372     suggest delete "an"

526     reference 52 is not in the text

Line 288/ 326

533     reference 55 is not in the text

Line 258

Table 1.  I would suggest somehow make the lines separating the different approaches more distinct, darker or thicker in some way.  In the column under "Performance" for polymorphisms of TGF-B indicates "no consistent association with acute" acute what? perhaps acute rejection?

Suggest overall Table 1 name heading to delete "their use"

Table 2.  I would suggest somehow make the lines separating the different approaches more distinct, darker or thicker in some way.  Second column "VIRTUUS panel".  Third column "Prediction of Acute Rejection".  Fourth column spelling of "independent" and "Indeed"

Suggest overall Table 2 name heading to delete "their use". 

Reviewer 2 Report

The paper entitled "Molecular markers of kidney transplantation outcome: current omics tools and future developments" unifies the knowledge of available biomarkers of kidney transplantation in an accessible way. The article presents a large amount of information and I therefore propose that it be accepted for publication with minor revisions:

1. The authors often use the term small cohorts, it is inaccurate - please supplement the information about study and control groups in the cited articles.

2. Information about the origin of the biomarker was missing in the text. Please add to the table or text data on whether the biomarker was detected in blood/urine or tissue. 

Author Response

Dear Reviewer,

Thank you for your helpful comments and inputs. We have amended the papers accordingly, and below we list the different changes in response to your comments.

The paper entitled "Molecular markers of kidney transplantation outcome: current omics tools and future developments" unifies the knowledge of available biomarkers of kidney transplantation in an accessible way. The article presents a large amount of information and I therefore propose that it be accepted for publication with minor revisions:

  1. The authors often use the term small cohorts, it is inaccurate - please supplement the information about study and control groups in the cited articles.
  2. Information about the origin of the biomarker was missing in the text. Please add to the table or text data on whether the biomarker was detected in blood/urine or tissue

Thank you for all your comments. We did the changes directly on the text. Indeed, we changed “small cohort” by “limited size study” and supplemented the information about study group in the form “n=…”. We added the information of biological matrix for all studies in the table 1 and 2.

This manuscript is a resubmission of an earlier submission. The following is a list of the peer review reports and author responses from that submission.